# MBD3 as a Potential Biomarker for Colon Cancer: Implications for Epithelial-Mesenchymal Transition (EMT) Pathways

**DOI:** 10.3390/cancers15123185

**Published:** 2023-06-14

**Authors:** Yuntao Ding, Huizhi Wang, Junqiang Liu, Han Jiang, Aihua Gong, Min Xu

**Affiliations:** 1Department of Gastroenterology, Affiliated Hospital of Jiangsu University, Jiangsu University, Zhenjiang 212000, China; 2222113004@stmail.ujs.edu.cn (Y.D.); 2112013011@stmail.ujs.edu.cn (H.W.); 2221613035@stmail.ujs.edu.cn (J.L.); 2221913006@stmail.ujs.edu.cn (H.J.); 2Hematological Disease Institute of Jiangsu University, Affiliated Hospital of Jiangsu University, Jiangsu University, Zhenjiang 212001, China

**Keywords:** colon cancer, EMT, bioinformatics, immunohistochemistry, biomarker

## Abstract

**Simple Summary:**

The tumor epithelial–mesenchymal transition (EMT) is a critical event in tumor pathogenesis and progression. While MBD3’s significant role in pancreatic cancer EMT has been established, its precise role in colon cancer remains unclear and requires further investigation. Pan-cancer analysis has revealed differential expression of MBD3 in various tumors, significantly associated with tumor occurrence, growth, and progression. Furthermore, analysis of single-cell sequencing and clinical data for colon cancer has revealed a negative correlation between MBD3 expression and clinical indicators such as survival prognosis. Functional enrichment analysis has confirmed the association between MBD3 and EMT in colon cancer. These findings demonstrate MBD3’s potential as a prognostic marker and therapeutic target for colon cancer.

**Abstract:**

The tumor EMT is a crucial event in tumor pathogenesis and progression. Previous research has established MBD3’s significant role in pancreatic cancer EMT. However, MBD3’s precise role in colon cancer remains unclear and warrants further investigation. Pan-cancer analysis revealed MBD3’s differential expression in various tumors and its significant association with tumor occurrence, growth, and progression. Moreover, analysis of single-cell sequencing and clinical data for colon cancer revealed MBD3 expression’s negative correlation with clinical indicators such as survival prognosis. Functional enrichment analysis confirmed the association between MBD3 and EMT in colon cancer. Pathological examinations, western blotting, and qRT-PCR in vitro and in vivo validated MBD3’s differential expression in colon cancer. Transwell, CCK-8, clone formation, and in vivo tumorigenesis experiments confirmed MBD3’s impact on migration, invasion, and proliferation. Our findings demonstrate MBD3 as a potential prognostic marker and therapeutic target for colon cancer.

## 1. Introduction

Colon adenocarcinoma (COAD) is a significant public health concern, ranking as the fourth most common cancer worldwide and exhibiting a worrisome increase in incidence rates [1,2]. Despite the proven efficacy of surgical intervention and adjuvant chemotherapy in treating COAD [3], the identification of novel therapeutic targets and robust prognostic markers remains a critical research priority. While the use of carcinoembryonic antigen (CEA) and carbohydrate antigen 19-9 (CA199) as biomarkers for COAD diagnosis and prognosis prediction is prevalent in clinical settings, their suboptimal performance in this regard has been well-documented [4].

Epithelial–mesenchymal transition (EMT) is a complex biological process that underlies the acquisition of mesenchymal features by epithelial cells, which is a critical step in tumor progression, invasion, metastasis, and drug resistance [5,6,7]. Given the importance of EMT in cancer biology, identifying key molecules that influence its development is of paramount importance. The methyl-CpG-binding domain (MBD) family of proteins, which are known to interact with methylated CpG dinucleotides, have been implicated in the pathogenesis of various diseases, including cancer [8]. MBD3, a member of the MBD gene family, has been shown to play a role in the development of several digestive tract tumors. In liver cancer, MBD3 has been shown to promote tumor cell growth, angiogenesis, and metastasis by inhibiting the tumor suppressor tissue factor pathway inhibitor 2 (TFPI2) [9]. In pancreatic cancer, MBD3 has been shown to inhibit EMT through the TGF-β/Smad signaling pathway [10] and stemness through the Hippo pathway [11]. However, the precise role of MBD3 in EMT in COAD remains unclear.

In this study, we utilized a multi-pronged approach, including cytological experiments, animal experiments, patient pathological sections, and various bioinformatics methods, to investigate the potential molecular mechanisms of MBD3 in the development and clinical prognosis of COAD. Specifically, we performed single-cell sequencing, a comprehensive analysis of MBD3 expression profiles, survival status, and potential molecular pathways in TCGA and GEO databases. We further validated differential expression in patient pathological sections and COAD cell lines and explored the effect of MBD3 on EMT in COAD cells through animal experiments and cytological experiments.

## 2. Materials and Methods

### 2.1. Data Download Gene Expression Analysis

We downloaded 33 kinds of tumor project STAR process RNAseq data and extracted the TPM format from the TCGA database (https://portal.gdc.cancer.gov, accessed on 12 December 2022). The relevant data of normal tissues and cells were downloaded from the Genotype-Tissue Expression (GTEx) database. Transcripts permillion reads (TPM) are used to standardize the HTSeq FPKM Level 3 data. R software v4.2.1 was used for statistical analysis, and the ggplot2 package was used for visualization. The Wilcoxon rank-sum test was used to detect the data of the two groups, and *p* < 0.05 was considered statistically significant (ns, *p* ≥ 0.05; *, *p* < 0.05; *, *p* < 0.01; *, *p* < 0.001).

### 2.2. The MSI Analysis and Gene Mutation Landscape of MBD3

We downloaded the harmonized pan-cancer dataset from the UCSC (https://xenabrowser.net/, accessed on 12 December 2022) database: TCGA Pan-Cancer (PANCAN, N = 10535, G = 60499); we further extracted ENSG00000071655 (MBD3) expression data in each sample, and further screened the sample source as Primary Blood Derived cancer-peripheral Blood, Primary Tumor samples. From a previous study (Landscape of Microsatellite Instability Across 39 Cancer Types, DOI:10.1200/PO.17.00073) [12], we integrated the MSI and gene expression data of the samples, and further applied log_2_(x + 0.001) transformation to each expression value. In addition, we downloaded software MuTect2 (https://portal.gdc.cancer.gov/, accessed on 12 December 2022) from GDC for processing all Simple level4 TCGA sample Nucleotide Variation dataset; we calculated MATH (Mutant-allele tumor) for each tumor using the inferHeterogeneity function of the R package maftools (version 2.2.10). We further applied a log_2_(x + 0.001) transformation to each expression value by integrating the TMB and gene expression data of the samples and excluded cancers with less than 3 samples in a single cancer, resulting in 37 cancer expression data. For the Variation dataset, we integrated the mutation data of the samples and obtained the protein domain information from the R package maftools (version v2.2.10). The full names and abbreviations of cancers are shown in Appendix A.

### 2.3. Single Cell Sequencing

CancerSEA (Yuan et al., 2019) is a specialized single-cell sequencing database that can provide different functional states of cancer cells at the single-cell level. The correlation between MBD3 expression and different tumor functions was analyzed based on single-cell sequencing data. The T-SNE plot shows the MBD3 expression profile of single cells in TCGA samples.

### 2.4. Survival Prognosis Analysis

Kaplan–Meier plots were used to assess the relationship between MBD3 expression and prognosis (OS) of cancers. Proportional-hazards hypothesis testing and fitted survival regression were performed with the survival package (version 3.3.1), and the results were visualized with the survminer package and the ggplot2 (version 3.3.6) package. The Log-rank test was used in the hypothesis test, and *p* < 0.05 is considered statistically significant.

### 2.5. Clinical Significance of MBD3 in COAD

Risk score, Calibration, Nomogram, and forest map were used for further clinical significance analysis of MBD3 in COAD. Risk score maps were visualized with the ggplot2 package (version 3.3.6). The survival package (version 3.3.1) was used for proportional hazards hypothesis testing and Cox regression analysis, and the rms package (version 6.3-0) was used for Calibration analysis and visualization. The survival package was used for proportional hazards hypothesis testing and Cox regression analysis, and the rms package was used to construct and visualize the nomogram correlation model. Forest map visualization was performed using ggplot2 (version 3.3.6).

### 2.6. Co-Expression Gene Analysis of MBD3 and Function Enrichment in COAD

We extracted the data of the corresponding molecules from the TCGA public database and divided them into a high-expression group and a low-expression group according to the expression of the corresponding molecules. The raw Counts matrix of the selected public data was analyzed using the DESeq2 package (version 1.36.0) following standard procedures. Using Pearson’s correlation coefficient, we also showed the correlation between MBD3 expression and the expression of the top 5 positively correlated genes and the top 5 negatively correlated genes using heat maps and lollipop plots. Functional enrichment between co-expressed genes and MBD3 in colon cancer was predicted by KEGG, GO and GSEA analysis. The above data were visualized using ggplot2 (version 3.3.6).

### 2.7. Cell Line and Cell Culture

Colon cancer cell lines SW620, SW480, CaCo2 and HCT116 were provided and maintained by the Central Laboratory of the Affiliated Hospital of Jiangsu University and the Institute of Basic Medicine, Jiangsu University School of Medicine. Colon cancer cell lines SW620, SW480, CaCo2, and HCT116 were cultured in DMEM (Hyclone, Beijing, China) supplemented with 10% fetal bovine serum, Gibco, Carlsbad, CA, USA) in 100 mg 1 penicillin in a 37 °C humidified incubator with 5% CO_2_ supply.

### 2.8. RNA Extraction and Real-Time PCR

Total RNA was extracted using Trizol (Invitrogen, Carlsbad, CA, USA), and for nude mouse tissues, tissue blocks were placed directly in a acetabulum with a small amount of liquid nitrogen, followed by rapid grinding. After the tissue was softened, a small amount of liquid nitrogen was added and then ground again and repeated three times. Trizol was added at 50 to 100 mg tissue/mL, transferred to a centrifuge tube, and the homogenate was thoroughly homogenized for about 1–2 min using an electric homogenizer. Reverse transcription was performed using the RevertAid first-strand cDNA Synthesis Kit (Thermo, Waltham, MA, USA) according to the manufacturer’s instructions. Quantitative real-time PCR was performed using iQ SYBR Premix Ex Taq Perfect Real Time from Bio-Rad Laboratories in 10 μL tubes, and SYBR was screened with DNA-specific fluorescent dye. Human U6 was chosen as the housekeeping gene. In this system, the primer pairs used to amplify the human UCA1 gene and human U6 were as follows in Appendix A. Primer sequences of EMT-related molecules were as follows in Appendix A. The samples were cycled under the following conditions: 95 °C for 3 min, 95 °C for 20 s, 56 °C for 20 s, and 72 °C for 30 s for 40 cycles. The relative expression of genes was calculated by the comparative CT method (ΔΔCT), and the fold enrichment was determined as follows: 2 − [ΔCT(sample) − ΔCT(calibrator)].

### 2.9. Cell Total Protein Extraction and Western-Blot

Cultured cells were rinsed with cold PBS and treated with RIPA lysis buffer at 4 °C for 10 min, followed by heating at 100 °C for 10 min, centrifugation at 14,000× *g*/min at 4 °C for 10 min, removal of the supernatant, and determination of protein concentration by the BCA assay. Each lane was loaded with about 20 mg of protein, separated by 10% SDS-PAGE and transferred to PVDF membranes. Membranes were blocked with 5% skim milk powder for 1 h at room temperature, followed by incubation with primary antibodies overnight at 4 °C and secondary antibodies for 1 h at room temperature. The liquid was uniformly dropped on the membrane from an appropriate amount of ECL, and the membrane surface was uniformly covered with color solution. The image was photographed and analyzed by chemiluminescence imaging analysis software.

### 2.10. Plasmid Construction, Transfection and Infection

The complete MBD3 sequence was amplified by RT–PCR using primers MBD3-all-F (5′-CGGAATTCCGATGGAGCGGAAGAGCCCGAGCG-3′) and MBD3-all-R (5′-GGGGTACCCCCTAGACGTGCTCCATCTCCGGGT-3′) from a cDNA library of PANC1 cells, then inserted into the expression Vector p3xFLAG-Myc-CMV-24 (Sigma, St. Louis, MO, USA). The sh-EGFP and sh-MBD3 plasmids were previously constructed in our laboratory (Sigma) and kept at the School of Basic Medical Sciences, Jiangsu University. Plasmids were transfected into colon cancer cells using Lipofectamine 2000 (Invitrogen) according to the manufacturer’s instructions. The specific sequences are shown in Appendix A. Methods for generating retroviruses encoding reprogramming factors and further infecting NPCs were referred to a previous paper (Pontes et al., 2014).

### 2.11. Transwell Migration and Invasion Assay

Transwell assays were performed using transwell inserts (Corning, Corning, New York, NY, USA) containing 8 mm permeable Wells according to the manufacturer’s protocol. Transfected SW620, SW480, CaCo2, and HCT116 cells were harvested, resuspended in serum-free medium, and transferred to 8 μm permeable wells (100,000 cells per well). The cells were then incubated with culture medium containing 10% FBS for 24 h before detection. The cells on the upper surface were scraped off, and the migrating cells on the lower surface were fixed and stained with 0.05% crystal violet for 30 min. Finally, five independent fields per transwell were counted, and the average number of cells per field is represented in the figure. To assess cell invasion, 100,000 cells were seeded in Matrigel-coated transwell inserts (BD Bioscience, Corning, NY, USA) in serum-free medium. Cells were then treated similarly to cell migration assays.

### 2.12. Cell Proliferation Assays

Cell proliferation was detected by cell-counting kit-8 (CCK-8, Beyotime Institute of Biotechnology, Shanghai, China). For the CCK-8 assay, 2 × 104 cells were seeded in 96-well plates for 24 h and transfected with Vector, Flag-MBD3, sh-EGFP, and sh-MBD3 in colon cancer cells. At 0, 1, 2, 3, 4, 5, and 6 days after transfection, 10 μL of cell-counting kit solution was added to each well; 96-well plates were incubated at 37 °C for 2 h, and absorbance values at each time point were measured at 450 nm using a microplate reader. All experiments were performed with at least three biological replicates.

### 2.13. Colony Formation Assay

Stable cell lines were collected, re-suspended in medium, transferred to 6-well plates (500 cells per well), and cultured for 10 to 14 days until large colonies appeared. Cells were fixed in 4% paraformaldehyde for 15 min and then stained with 0.05% crystal violet for 30 min to count the number of colonies.

### 2.14. Xenograft Mouse Model

The protocol was approved by the Institutional Animal Care and Use Committee of Jiangsu University, Zhenjiang, China. SW620 cells (2.0 × 10^6^ cells/site) stably transfected with shEGFP and shMBD3.CaCO2 cell lines stably transfected with Flag-Vector and Flag-MBD3 were subcutaneously injected into 5-week-old BALB/c nude mice (Shanghai SLAC Laboratory Animal Co., Ltd., Shanghai, China) to generate xenografts. There are four female mice in each group. Tumor volume was measured weekly after injection and calculated using the formula: length × width × height × π/6.

### 2.15. Pathological Sample Collection

A total of 3 samples of colon cancer tissues and their matched paracancerous tissues were collected between March 2023 and April 2023 at the Affiliated Hospital of Jiangsu University. This study was approved by the medical ethics committees of Affiliated Hospital of Jiangsu University and was conducted in line with the Declaration of Helsinki.

### 2.16. Immunohistochemistry

Immunohistochemistry (IHC) staining was carried out as previously described (DeRycke et al., 2009). Tumor tissues and paracancerous tissues were fixed in 10% formalin, paraffin-embedded, sliced into 4~6 μm sections, and placed onto slides. After deparaffinization, rehydration and microwave antigen retrieval, the slides were incubated with MBD3 (Proteintech Cat No. 14258-1-AP) antibody at 1:800 dilution at 4 °C overnight. Then, the slides were incubated with secondary antibody at room temperature for 30 min and stained with DAB substrate, followed by hematoxylin counterstaining.

### 2.17. Statistical Analyses

All data are presented as mean ± standard deviation, from at least three independent experiments. The *t* test was used for comparison between two groups, and one-way analysis of variance was used for comparison between multiple groups. Kaplan–Meier survival analysis was performed using the log-rank test. A *p* value of less than 0.05 was considered statistically significant.

## 3. Results

### 3.1. Gene expression Analysis

We explored the relative expression of MBD3 in cancer tissues and adjacent tissues. Pan-cancer analysis showed that it was highly expressed in a variety of cancers, such as CHOL, LIHC, LUAD, COAD, etc. (Figure 1A), which was further verified by pan-cancer analysis of control samples (Figure 1C). The expression of MBD3 in colon cancer samples was significantly higher than in normal colon tissues (*p* < 0.001) (Figure 1B). In addition, control samples supported this conclusion (*p* < 0.001) (Figure 1D). MBD3 expression profile at the single-cell level of COAD was also shown by T-SNE plots (Figure 1E).

### 3.2. Prognostic and Clinical Significance of MBD3 in Colon Cancer

We explored the effect of MBD3 on colon cancer prognosis using KM curves, which showed that low MBD3 expression resulted in a better prognosis, while high MBD3 expression resulted in a poor prognosis (Figure 2A), and risk factor plots further confirmed that MBD3 high expression was associated with poor prognosis (Figure 2B). To further discuss the clinical significance of MBD3 in colon cancer, we used calibration analysis to predict the association of MBD3 expression with 1, 2, and 3-year prognosis in colon cancer patients. At the same time, we used MBD3 as one of the independent OS factors to construct a prognostic calibration curve for predicting the prognosis of colon cancer patients, and the prediction results suggested that the fit was good, and the survival rate was consistent with the prediction results of the model (Figure 2C,D). Finally, univariate and multivariate Cox regression analyses were used to find the prognostic factors. Univariate prognostic analysis showed that MBD3 expression was significantly associated with T3 stage (HR = 2.576, 95% CI = 1.183–5.612, *p* = 0.017), T4 stage (HR = 7.021, 95% CI = 2.993–16.473, *p* < 0.001), N1 stage (HR = 1.681, 95% CI = 1.019–2.771, *p* = 0.042), N2 stage (HR = 4.051, 95% CI = 2.593–6.329, *p* < 0.001), M1 stage (HR = 4.193, 95% CI = 2.683–6.554, *p* < 0.001), CR (HR = 0.111, 95%CI = 0.057 −0.214, *p* < 0.001), having lymphatic invasion (HR = 2.450, 95% CI = 1.614–3.720, *p* < 0.001) and age > 65 years (HR = 1.610, 95% CI = 1.052–2.463, *p* = 0.028). In a multivariate prognostic analysis, MBD3 expression was associated with CR (HR = 0.120, 95% CI = 0.040–0.360, *p* < 0.001) (Figure 2E,F).

### 3.3. The MIS Analysis of MBD3 and Gene Mutation Landscape

The lollipops of the gene mutation landscape suggested that the mutation sites of MBD3 were mostly located in the methyl-CPG binding domain and were mainly related to MeCP2 MBD and MBDC (Figure 3A). To investigate the effect of MBD3 on tumor microsatellite instability (MSI), we calculated their Pearson correlation in each tumor, and we observed a significant correlation in 13 tumors, Among them, GBMLGG (N = 657) (R = 0.0776721765681662, *p* = 0.0465787995676234) and LUAD (N = 511) (R = 0.219060376536456, P = 5.7132985511611) were significantly positively correlated (9 × 10^−7^), KIPAN (R = 0.0964745324112317, P = 0.0113475547444957), PRAD (N = 495) (R = 0.234547294485744, P = 1.29911971978673 × 10^−7^), UCE C (N = 180) (R = 0.246779767475385, P = 0.000838882444752513), HNSC (N = 500) (R = 0.140955507023987, P = 0.00157898229854812), LUSC(N = 490) (R = 0.223985227211001, P = 5.46440716199723 × 10^−7^), TGCT (N = 148) (R = 0.171567605892632, P = 0.0370694378555342), DLBC (N = 47) (R = 0.53171230) 1125292, P = 0.000120027065256395), which was significantly negatively correlated in four tumors: for example, COAD (N = 285) (R = −0.152035894437063, P = 0.010159865013793), COADREAD (N = 374) (R = −0.126434656888683, P = 0.0144142145160222), STES (r = −0.152035894437063, P = 0.010159865013793) N = 592) (R = −0.164712782169241, P = 0.0000565653082653195), STAD (N = 412) (R = −0.118854942454545, P = 0.015790433741618) (Figure 3B). To investigate the effect of MBD3 on mutant-allele tumor heterogeneity (MATH), we calculated their Pearson correlation in each tumor. We observed significant correlations in five tumors, including a significant positive correlation in one tumor, and a significant positive correlation in one tumor. For example, STES (N = 589) (R = 0.156393954823199, P = 0.000140244272553858) was significantly negatively correlated in four tumors. For example, KIRP (N = 279) (R = −0.134531328308061, P = 0.0246238619162398), KIPAN (N = 679) (R = −0.116894739656509, P = 0.00228192599956915), UCEC (N = 175) (R = −0.338969011255145, P = 0.00000446255622106516), UCS (N = 57) (R = −0.301762259882098, P = 0.0225339517974099) (Figure 3C).

### 3.4. The Expression Pattern of MBD3 at Single-Cell Levels

Single-cell analysis can more accurately analyze the underlying molecular mechanisms of genes at the single-cell level (He et al., 2021; Li et al., 2021). Figure 4A suggests that MBD3 participates in a variety of biological processes in tumors, such as EMT, Hypoxia, and invasion. In colon cancer, MBD3 was positively correlated with Differentiation, Angiogenesis, Inflammation, Hypoxia, and Apoptosis, and negatively correlated with CellCycle and DNAdamage (Figure 4B).

### 3.5. Analysis of Related Differentially Expressed Genes of MBD3 and Function Enrichment in COAD

We used the Dseq2 R package to analyze the differentially expressed genes (DEGs) of MBD3 in COAD. The results showed that there were 4824 differentially expressed genes between the MBD3 high-expression group and the MBD3 low-expression group, including 72 up-regulated genes and 4752 down-regulated genes (*p* < 0.05, |Log2 − FC| > 1.5) (Figure 5A). The relationship between the top five highly expressed DEGs and the top five low-expressed DEGs (including MIR8075 HBZ DLK1 APOA4 AP001531.1 MIR3609 RNY1 RNU1−11P RNU1−88P RN7SKP203) and MBD3 is shown in Figure 4D,E. In GO enrichment analysis, the DEGs-related functions of MBD3 were mainly enriched in BP for detection of chemical stimulus involved in sensory perception, nucleosome assembly, nucleosome organization related to DNA replication-dependent chromatin assembly. Nucleosome assembly was especially related to EMT (Figure 4F). Enriched in KEGG function. MBD3 and DEGs were mainly enriched in: Systemic lupus erythematosus, Alcoholism, Neutrophil extracellular trap formation, Olfactory transduction and Taste transduction (Figure 4G), GSEA functional enrichment suggests that they are involved in Transport of Small Molecules and Hemostasis (Figure 4H).

### 3.6. The Expression of MBD3 in COAD Cell Lines and Tissue

QRT-PCR and Western-blot showed that MBD3 was differentially expressed in colon cancer cell lines, among which the expression level was the highest in the SW620 cell line, the lowest in the CaCO2 cell line, and the second and third in the HCT116 and SW480 cell lines, respectively (Figure 6A). Subsequently, the efficiency of knockdown and overexpression of sh-MBD3 and Flag-MBD3 was verified, and the results showed that the relative expression of MBD3 in colon cancer cells transfected with sh-MBD3 was significantly lower than in the control group (Figure 6B). However, the expression level of MBD3 in colon cancer cells transfected with Flag-MBD3 plasmid was significantly higher than in the control group (Figure 6C). Figure 5A–C showed IHC sections of colon cancer tissues and adjacent colon cancer tissues from three elderly male patients at 20× and 40× magnification, respectively. The expression of MBD3 was up-regulated. Representative images are presented in Figure 5.

### 3.7. Effect of MBD3 on the Migration and Invasion Ability of COAD Cells

We used a transwell assay to verify the cell migration and invasion ability. Knockdown of MBD3 in SW620 and HCT116 cell lines and setting up a control group showed that the cell migration ability of the sh-MBD3 group was significantly weaker than that of the sh-EGFP group (Figure 6G,H). However, in SW840 and CaCO2 cell lines overexpressing MBD3, the migration ability of the Flag-MBD3 group was significantly stronger than that of Vector group (Figure 6I,J). Similarly, knockdown of MBD3 and setting up a control group in SW620 and HCT116 cell lines suggested that the cell invasion ability of the sh-MBD3 group was significantly weaker than that of the sh-EGFP group (Figure 6K,L). However, in SW840 and CaCO2 cell lines overexpressing MBD3, the invasion ability of the Flag-MBD3 group was significantly stronger than that of Vector group (Figure 6M,N).

### 3.8. Effect of MBD3 on Colon Cancer Proliferation

CCK-8 experiment was used to further verify the effect of MBD3 on cell proliferation. Knockdown of MBD3 in SW620 and HCT116 cell lines and the establishment of a control group indicated that the cell proliferation ability of sh-MBD3 group was significantly weaker than that of the sh-EGFP group (Figure 7A,B). However, in SW840 and CaCO2 cell lines overexpressing MBD3, the proliferation ability of the Flag-MBD3 group was significantly stronger than that of the Vector group (Figure 7C,D). In addition, knockdown expression of MBD3 inSW620 and HCT116 cell inhibition of ed-proliferation and colony formation (Figure 7E,F) and overexpression of MBD3 in CaCo2 and SW480 cells promoted proliferation and colony formation (Figure 7G,H). An examination of the in vivo effect of the MBD3 revealed slow tumor growth in mice injected with sh-MBD3-transfected SW620 cells compared to control cells (Figure 7I,J). Additionally, Flag-MBD3-transfected CaCo2 cells generated bigger tumors than control cells (Figure 7K,L). The differential expression of EMT-related molecules between MBD3 knockdown SW620 cells and MBD3 overexpression CaCO2 cells obtained from nude mice tissues is shown in Appendix A. These results indicate that the MBD3 plays an oncogenic role during COAD progression.

## 4. Discussion

The methyl-CpG-binding domain (MBD) protein family has been implicated in a variety of biological processes, including tumorigenesis [8,13,14]. Specifically, MBD2 has been shown to promote the progression and poor prognosis of renal cell carcinoma [15], while MBD4 has been associated with cervical cancer polymorphism [16]. MBD3 has been demonstrated to promote the metastasis and growth of various digestive tract tumors, and, in liver cancer, it can enhance the growth, angiogenesis, and metastasis of tumor cells by inhibiting the tissue factor pathway inhibitor 2 (TFPI2) [9]. In pancreatic cancer, MBD3 can inhibit the epithelial–mesenchymal transition (EMT) process through TGF-β/Smad signaling transduction [10] and suppress the stemness of pancreatic cancer cells through the Hippo pathway [11]. However, the role of MBD3 in colon cancer remains to be fully elucidated.

Colon cancer (COAD) is the fourth most common cancer worldwide, with an increasing incidence rate [1]. Early diagnosis of COAD currently relies on invasive techniques such as colonoscopy [17], and while molecular biology markers such as CEA and CA199 have been widely utilized, the need for more sensitive molecular markers remains urgent [18,19]. Although surgery remains the primary treatment for rectal cancer [20], new targeted therapies such as immunotherapy combined with anti-angiogenic drugs [21] and adjuvant chemotherapy [22] have improved patient prognosis. Thus, identifying novel therapeutic targets for COAD is of the utmost importance.

Based on the findings presented above, it is believed that MBD3 can facilitate the metastasis and growth of tumors through EMT, and may thus represent a promising biological marker and therapeutic target. To further explore the clinical significance of MBD3, we conducted an analysis of its expression differences across various tumors, with particular focus on colon cancer, utilizing a combination of TCGA and GTEx databases. We investigated its potential clinical relevance through a range of analytical techniques, including KM curves, hazard factor maps, nomograms, calibration, and univariate and multivariate COX regression analysis. The gene mutation landscape, tumor microsatellite instability, and t-SNE map of single-cell sequencing were also utilized to provide further insight into the role of MBD3 in tumor biology. Our results suggest that MBD3 has the potential to serve as a novel biological marker. Through functional enrichment analysis of co-expressed genes of MBD3 in colon cancer, we identified its involvement in multiple pathways of colon cancer biological processes and its association with EMT.

Colon cancer is a significant cause of mortality worldwide [23], with tumor metastasis being a leading cause of death [24]. Metastasis is the end product of a multistep cell-biological process of the invasion–metastasis cascade [25] and remains the principal cause of cancer death [26]; in colon cancer, tumor metastases such as liver metastases lead to poor prognosis [27]. Molecules such as E-cadherin and Snail have been shown to be associated with tumor metastasis, and they are key molecules in the EMT process [28,29]. EMT involved in fundamental processes in embryonic development and tissue repair and has been identified as a major factor in promoting cancer cell metastasis [30,31]. It has been confirmed in various tumors, including breast and gastric cancer [32,33]. More key molecules have been shown to affect the molecular mechanisms of EMT. For example, TGF-β1 can induce matrix POSTN to promote the migration and invasion of ovarian cancer [34], while the loss of PRC2 can affect the progression of prostate cancer [35]. In ovarian cancer, the Wnt/β-catenin axis has also been shown to affect the EMT process [2]. However, the role of MBD3 as a key molecule affecting EMT and metastasis in colon cancer has yet to be fully explored.

To address this gap in knowledge, we conducted an analysis of MBD3 using online databases TCGA and GETx and verified the differential expression of MBD3 in vivo and in vitro through the immunohistochemistry of patient pathological slices and total RNA qtPCR of cell lines. We subsequently verified the effect of MBD3 on tumor growth in animal models through nude mouse tumorigenesis experiments and further confirmed its impact on colon cancer migration, invasion, and proliferation through transwell and CCK8 experiments in colon cancer cell lines. However, the specific molecular mechanism of MBD3 on colon cancer EMT requires further investigation.

Despite the contributions of this study, there are some limitations to be noted. Firstly, the analysis was limited to the use of online databases TCGA and GETx, and the pathological slices obtained were also limited. Secondly, further investigation is required to clarify the specific molecular mechanism of MBD3’s effect on EMT, through techniques such as western blotting. Additionally, pattern animals that meet the necessary conditions will need to be constructed to further elucidate the mechanism of MBD3 in colon cancer.

In summary, this study has identified the potential role of MBD3 in colon cancer through bioinformatics and has evaluated its significance through a range of analytical techniques. The findings suggest that MBD3 plays an important role in multiple aspects of colon cancer, especially the EMT process, which promotes metastases and leads to poor prognoses, and has potential biological value as a novel therapeutic target.

## 5. Conclusions

The clinical relevance of MBD3 in colon cancer was analyzed by analytical techniques to identify its involvement in colon cancer pathways and EMT. MBD3 is expected to be a therapeutic target for colon cancer.

## Figures and Tables

**Figure 1 cancers-15-03185-f001:**
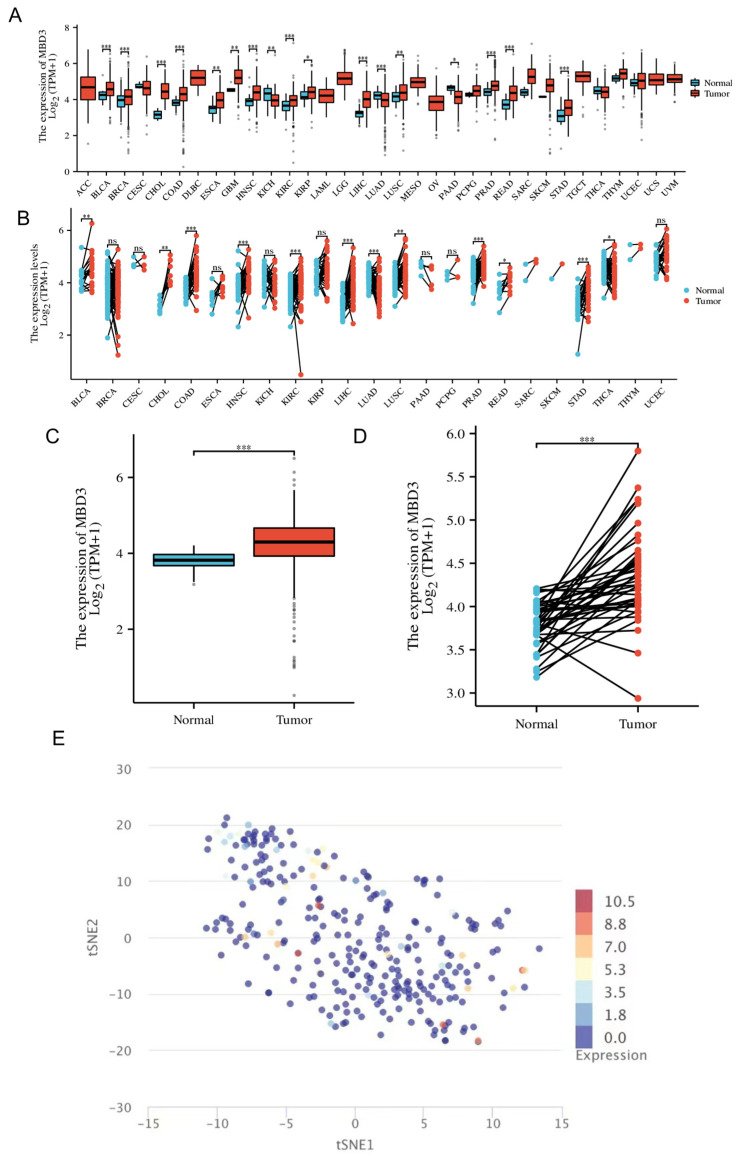
The relative expression of MBD3 in pan-cancer and COAD. (**A**,**B**) Different types of tumors compared with normal tissues in TCGA and GTEx databases. (**C**,**D**) Relative expression of MBD3 in colon cancer and colon cells. (**E**) Relative expression of MBD3 in single cells of COAD (* *p* < 0.05, ** *p* < 0.01, *** *p* < 0.001).

**Figure 2 cancers-15-03185-f002:**
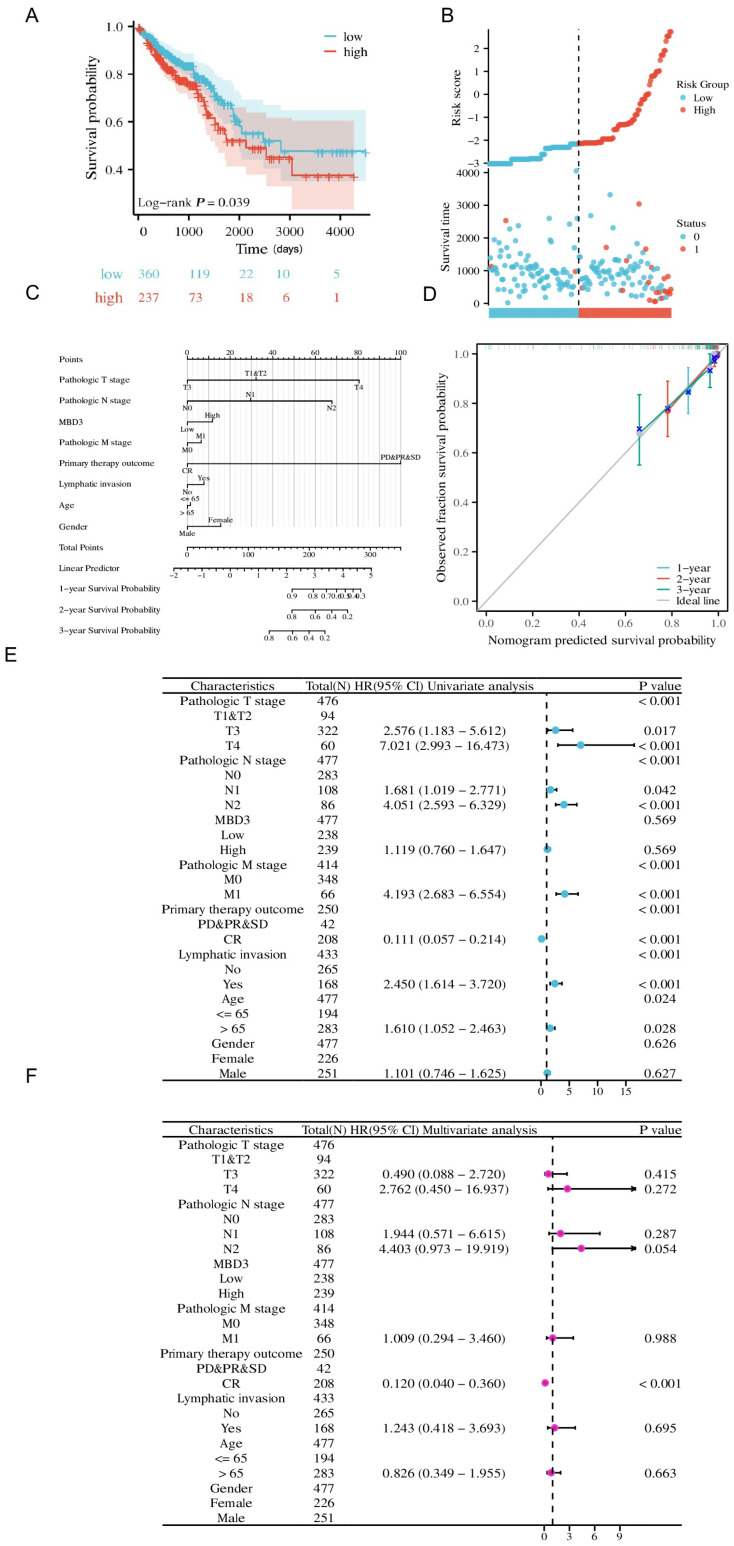
Clinical indicators of MBD3 in colon cancer. (**A**) Overall survival analyses of MBD3 in COAD. (**B**) Risk score of MBD3 in COAD. (**C**,**D**) Nomogram and calibration curves for prediction of one-, two-, and three-year overall survival rates of patients with COAD with high expression of MBD3. (**E**) Prognostic values of MBD3 expression by univariate analysis in COAD. (**F**) Prognostic values of MBD3 expression by multivariate analysis in COAD.

**Figure 3 cancers-15-03185-f003:**
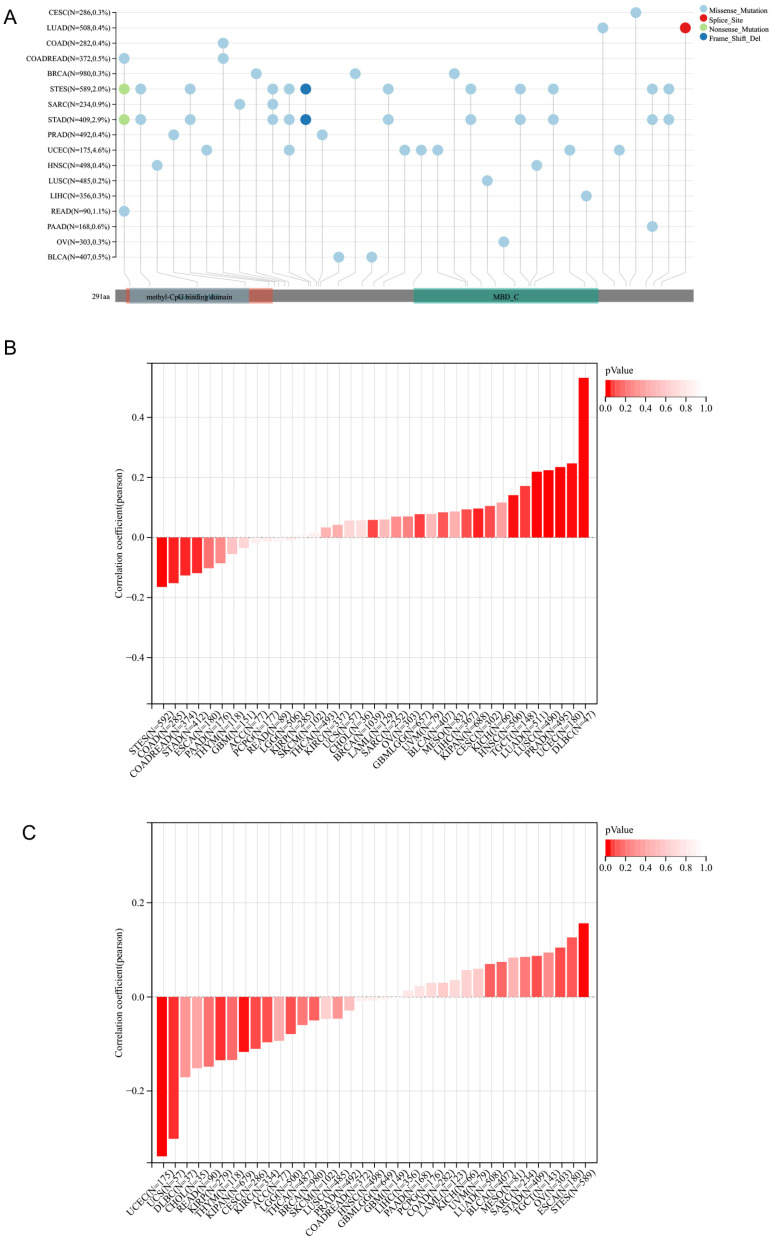
The immune mutational landscape, MSI and MATH of MBD3 in pan-cancer. (**A**) Immune mutational landscape of MBD3 in pan-cancer. (**B**,**C**) MSI and MATH of MBD3 in pan-cancer.

**Figure 4 cancers-15-03185-f004:**
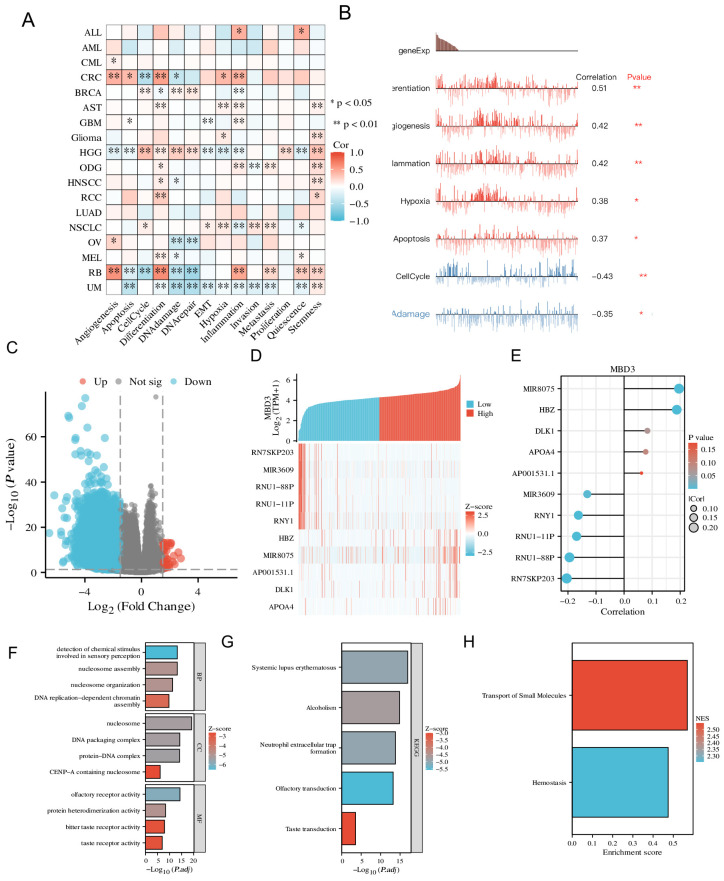
Function enrichment of MBD3 in single cells and DEGs. (**A**,**B**) Relationship between MBD3 expression and different functional states in tumors explored by the CancerSEA tool. * *p* < 0.05, ** *p* < 0.01. (**C**–**E**) Volcano plot of logFc > 1.5 and the top ten related differentially expressed genes of MBD3 from the TCGA database. (**F**–**H**) KEGG, GO and GSEA analysis of MBD3 and their co-expression genes.

**Figure 5 cancers-15-03185-f005:**
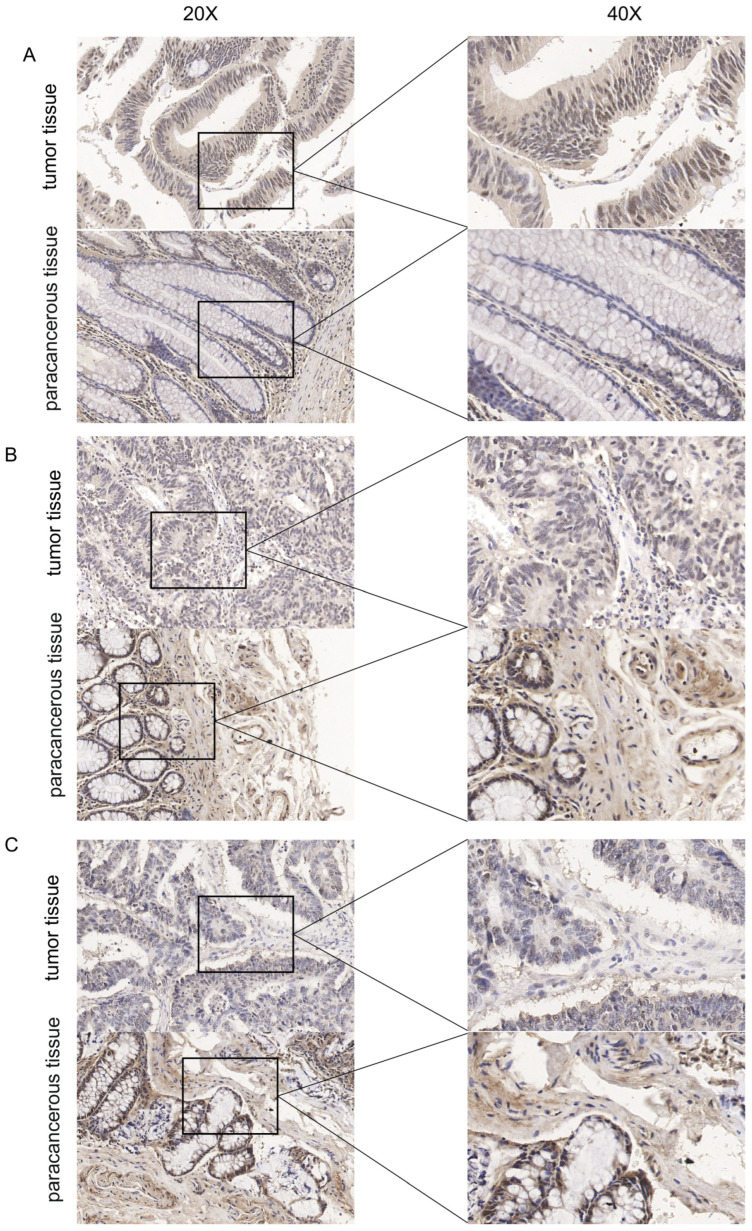
Representative images of MBD3 expression in COAD tissues and their matched paracancerous tissues. Original magnifications 20× and 40× (inset panels). (**A**–**C**) were representative images.

**Figure 6 cancers-15-03185-f006:**
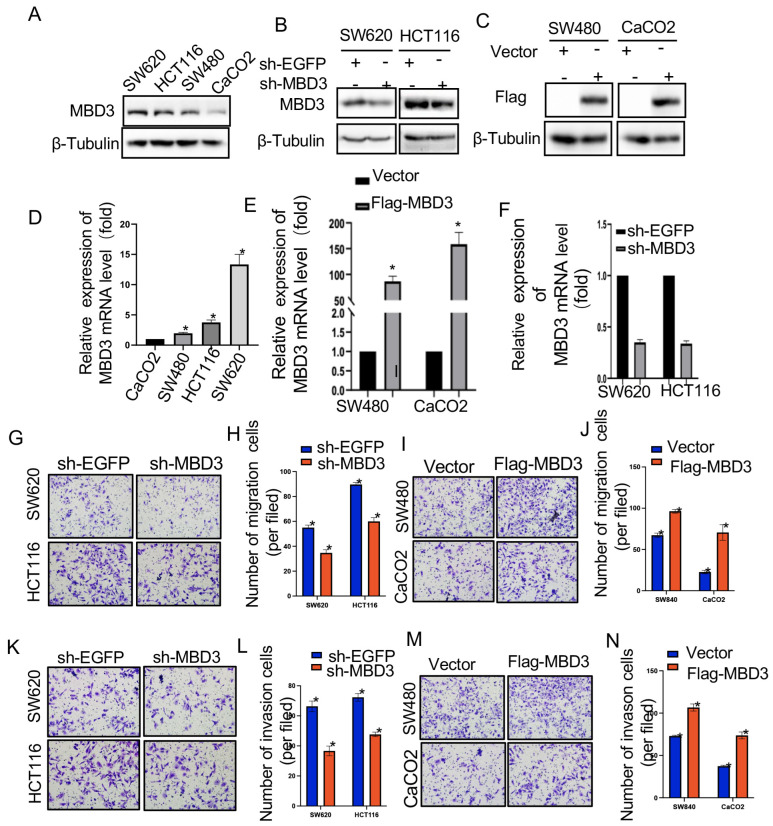
Cytological experiments regarding MBD3 (ns, (**D**) * *p* ≥ 0.05; (**E**) * *p* < 0.05; (**H**,**L**) * *p* < 0.01; (**J**,**N**) * *p* < 0.001). (**A**,**D**) The relative expression of MBD3 in COAD cell lines. (**B**,**C**,**F**) Plasmid transfection efficiency of MBD3 in COAD cell lines. (**G**) Cell migration assay of MBD3 in SW620, HCT116, SW480 and CaCO2. (**H**) Cell invasion assay of MBD3 in SW620, HCT116, SW480 and CaCO2. (**I**–**L**) show enhanced ability of MBD3 in COAD.

**Figure 7 cancers-15-03185-f007:**
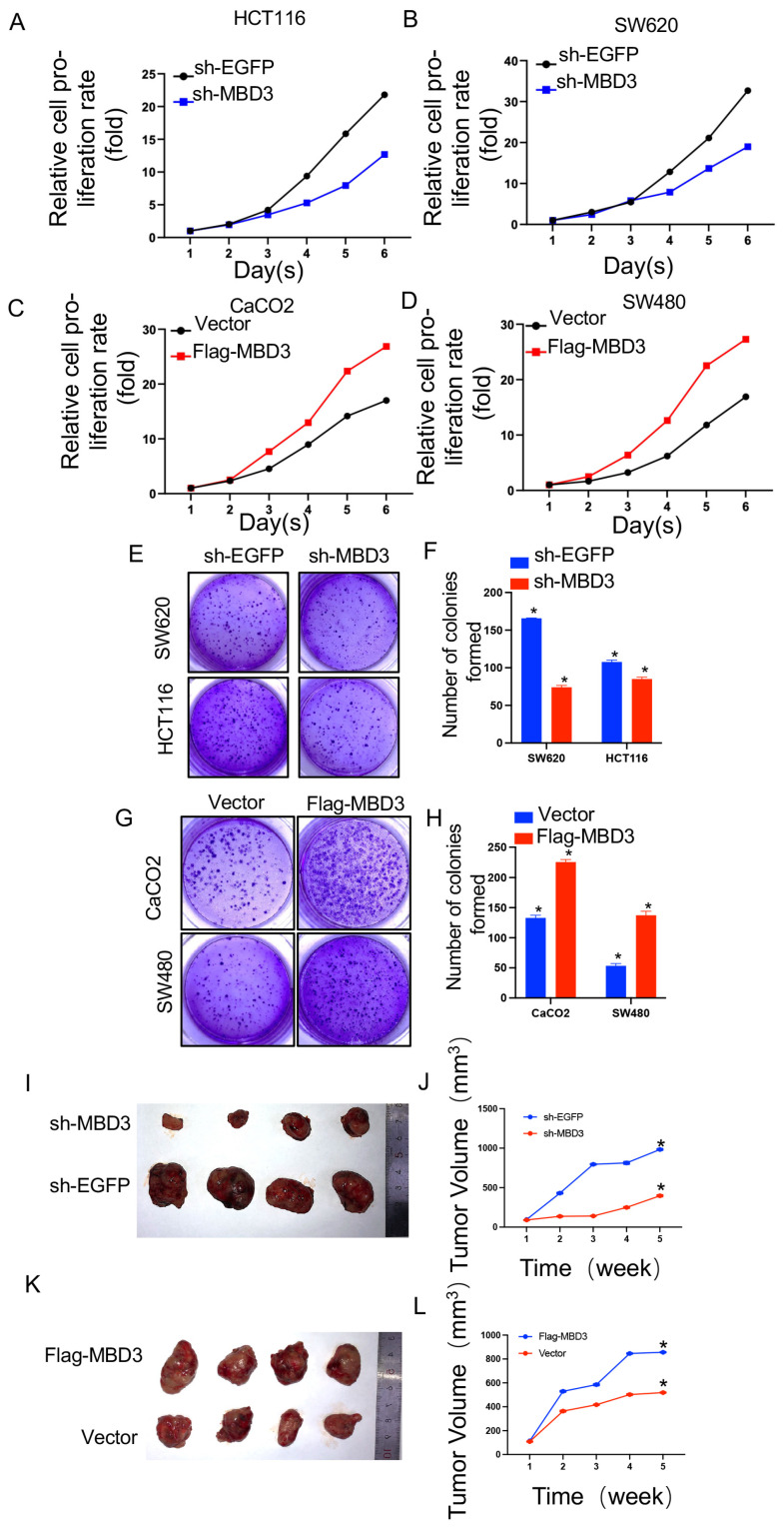
Effect of MBD3 on colon cancer proliferation and tumor formation experiment in nude mice. (**A**–**D**) Proliferative capacity of MBD3 in COAD. (**E**–**H**) Clone formation assay of MBD3 in COAD. (**I**–**L**) Tumor formation in nude mice (* *p* < 0.05).

## Data Availability

The data presented in this study are openly available in TCGA database https://www.cancer.gov/ccg/research/genome-sequencing/tcga (accessed on 12 December 2022).

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
