# Peer review of "MBD3 as a Potential Biomarker for Colon Cancer: Implications for Epithelial-Mesenchymal Transition (EMT) Pathways"

_cancers, 2023, doi:10.3390/cancers15123185_

Round 1
Reviewer 1 Report
In this article, the authors suggest that MBD3 could be a potential prognostic biomarker of colon cancer. They also implicate the role for epithelial-mesenchymal transition (EMT) pathways.
General comment:
The study is comprehensive and it includes bioinformatic and experimental analyses. The limitations of the study are fairly presented.
Specific point:
Since the authors have an access to the cells and to the tumors from nude mice with shRNA for MBD3, would it be possible to test any of the EMT markers in those systems (either by WB, qPCR or IHC)? This would strengthen the implication of EMT that is stated in the title of this manuscript.
Minor points:
Title: Instead of 'promising', a suggestion would be to write 'potential'
The third sentence in Introduction (lines 39-41) would better fit as a second sentence. The second sentence (lines 36-38) would better fit as a third sentence.
Section 2.14. Xenograft mouse model: It would be good to write how many mice per group were used.
Figure 2 is blurred and the font size could be increased. Figure 2a should indicate time unit (days).
Figure 6 is placed before Figure 5.
Figure 5: It should be written what panels A, B and C are and what COAD tissues and their matched paracancerous tissues are.
Figure 7A-C: X and y-axes should be explained.
Minor corrections needed.
Author Response
Dear Editor,
Thank you for your comments concerning our manuscript entitled “MBD3 as a Potential Biomarker for Colon Cancer: Implications for Epithelial-Mesenchymal Transition (EMT) Pathways” (ID: cancers-2424675). These comments were all valuable and very helpful for revising and improving our manuscript, as well as guiding the significance of our research. We have studied the comments carefully and have made corrections that we hope meet with approval. The revised portions are marked in red in the manuscript.
The main corrections in the manuscript and the responses to the editor’s comments are as follows.
- Comment:Title: Instead of 'promising', a suggestion would be to write 'potential’
Response: I feel sorry and thank you for your friendly advice. We have We have carefully revised the title of the manuscript as per your suggestion.
- Comment: The third sentence in Introduction (lines 39-41) would better fit as a second sentence. The second sentence (lines 36-38) would better fit as a third sentence.
Response: I feel sorry and thank you for your friendly advice. we have restructured the language to enhance the overall logical flow, making it more coherent and easier to follow.
- Comment: Section 2.14. Xenograft mouse model: It would be good to write how many mice per group were used.
Response: Based on your suggestions, we have made significant improvements to the methods section.
- Comment: Figure 2 is blurred and the font size could be increased. Figure 2a should indicate time unit (days).
Response: I feel sorry and thank you for your friendly advice. Based on your suggestions, we have add days in the figure.
- Comment: Figure 6 is placed before Figure 5.
Response: I feel sorry to make the mistake and thank you for your friendly advice. We have arranged the pictures in the correct order.
6 and 7. Comment: Figure 5: It should be written what panels A, B and C are and what COAD tissues and their matched paracancerous tissues are. And Figure 7A-C: X and y-axes should be explained.
Response: Your suggestion is quite useful for our study. We have also addressed the missing figure legends by replacing them with appropriate ones, ensuring clarity and accuracy in the presentation of our findings.
- Comment: Since the authors have an access to the cells and to the tumors from nude mice with shRNA for MBD3, would it be possible to test any of the EMT markers in those systems (either by WB, qPCR or IHC)? This would strengthen the implication of EMT that is stated in the title of this manuscript.
Response: We have conducted additional bioinformatics analyses, as you suggested, and have identified relevant molecules. However, we acknowledge that further research is required to fully investigate and understand their significance. We plan to incorporate these findings in our future studies.
If there are any other modifications that are needed, we would be happy to make further changes, and we appreciate your help. Thank you very much for your help.
Prof. Dr. Min Xu
Medical School
The affiliated hospital of Jiangsu University, ZhenJiang 212000, China
Email: peterxu1974 @163.com
Jun.03, 2023
Reviewer 2 Report
The authors extracted the significance of MBD3 as a biomarker in colorectal cancer from the results obtained from the database; the importance of MBD3 in EMT (EPithelial\Mesenchymal Transition) in pancreatic cancer has already been reported, but for the first time in colorectal cancer.
The database has been well analyzed and subsequently validated in cell lines and animal models, which are well documented and carefully tested.
The material, methods, and figures need to be more detailed. sh-EGPF, for example, needs to be detailed. In addition, I would like to see more detailed explanations of each panel in the figure descriptions ・・・・, especially in the results of experiments on cell lines, etc.
Author Response
Dear Editor,
Thank you for your comments concerning our manuscript entitled “MBD3 as a Potential Biomarker for Colon Cancer: Implications for Epithelial-Mesenchymal Transition (EMT) Pathways” (ID: cancers-2424675). These comments were all valuable and very helpful for revising and improving our manuscript, as well as guiding the significance of our research. We have studied the comments carefully and have made corrections that we hope meet with approval. The revised portions are marked in red in the manuscript.
The main corrections in the manuscript and the responses to the editor’s comments are as follows.
- Comment:The material, methods, and figures need to be more detailed. sh-EGPF, for example, needs to be detailed. In addition, I would like to see more detailed explanations of each panel in the figure descriptions ・・・・, especially in the results of experiments on cell lines, etc.
Response: Your suggestion is quite useful for our study. Expansion of Methods: We have expanded the methods section by providing additional details and elaborating on experimental procedures that were previously too concise. This will ensure that readers have a clear understanding of the methodology employed in our study.
If there are any other modifications that are needed, we would be happy to make further changes, and we appreciate your help. Thank you very much for your help.
Prof. Dr. Min Xu
Medical School
The affiliated hospital of Jiangsu University, ZhenJiang 212000, China
Email: peterxu1974 @163.com
Jun.03, 2023
Reviewer 3 Report
The research demonstrates MBD3 roles in colon cancer in terms of epithelial-mesenchymal transition.
1. Figure 5 is missing.
2. 2.8 RNA extraction and real-time PCR section may be revised to have a table of sequence of primers.
3. Conclusion may be expanded to emphasize the results.
Description of version of R package may be checked in 2.2. The MSI Analysis and Gene Mutation Landscape of MBD3.
Author Response
Dear Editor,
Thank you for your comments concerning our manuscript entitled “MBD3 as a Potential Biomarker for Colon Cancer: Implications for Epithelial-Mesenchymal Transition (EMT) Pathways” (ID: cancers-2424675). These comments were all valuable and very helpful for revising and improving our manuscript, as well as guiding the significance of our research. We have studied the comments carefully and have made corrections that we hope meet with approval. The revised portions are marked in red in the manuscript.
The main corrections in the manuscript and the responses to the editor’s comments are as follows.
- Comment:1. Figure 5 is missing.
Response: We apologize for the error in the placement of the images. We have rectified this issue by reformatting the manuscript, ensuring that the figures are correctly positioned and aligned with the corresponding text.
- Comment:2.8 RNA extraction and real-time PCR section may be revised to have a table of sequence of primers.
Response: In response to your suggestion, we have uploaded the relevant sequences as Supplementary Tables S2 and S3. Additionally, we have made the necessary modifications to the manuscript to reference these tables appropriately.
- Comment:Conclusion may be expanded to emphasize the results.
Response: We have expanded the discussion section to provide further elucidation on the relationship between MBD3, tumor metastasis, and the epithelial-mesenchymal transition (EMT) process. By incorporating additional information and referencing relevant literature, we have sought to strengthen the understanding of these connections and their implications in our study.
- Comment:Description of version of R package may be checked in 2.2. The MSI Analysis and Gene Mutation Landscape of MBD3.
Response: We apologize for the error in such palce, We have correct this error.
If there are any other modifications that are needed, we would be happy to make further changes, and we appreciate your help. Thank you very much for your help.
Prof. Dr. Min Xu
Medical School
The affiliated hospital of Jiangsu University, ZhenJiang 212000, China
Email: peterxu1974 @163.com
Jun.03, 2023